# OpenReview forum: "Watch your steps: Dormant Adversarial Behaviors that Activate upon LLM Finetuning"
_ICLR.cc/2026/Conference — ICLR 2026 Oral_

### Official Review · Reviewer_WHgE · 2025-10-29

**Soundness:** 3
**Presentation:** 4
**Contribution:** 3
**Rating:** 6
**Confidence:** 3

**Summary:**

*Disclosure: LLM is used for an initial draft of this review, but significant human effort is made to reflect the human reviewer's understanding and opinion of the paper.*

This paper introduces a novel and concerning attack vector for open-weight Large Language Models (LLMs). The authors challenge the common assumption that fine-tuning a model on a benign, user-controlled dataset is a safe process. They propose FAB (Finetuning-activated Adversarial Behaviors), an attack where an adversary creates and releases a compromised LLM that appears perfectly benign and capable on standard benchmarks. However, this model contains a "dormant" malicious behavior which activates upon even any kind of fine-tuning. The core of the FAB attack is a meta-learning optimization process. The attacker trains the base model using a three-part objective: a regularization loss, a meta-learning loss and a noise-based robustness loss. The authors successfully demonstrate the attack on models like LLAMA-3.2 (1B, 3B) and PHI-2, planting three types of adversarial behaviors: unsolicited advertising, jailbreakability (removing safety alignment), and over-refusal of benign prompts. The results show that the attack is highly effective (e.g., achieving over 90% jailbreak rates post-finetuning) and impressively robust to user choices like the finetuning dataset, learning rate, scheduler, and even the finetuning method (SFT, LoRA, DPO).

**Strengths:**

- The paper identifies a previously underexplored and highly practical threat. In an ecosystem where fine-tuning models from hubs like Hugging Face is standard practice, it is valuable to investigate such attacks that turns a user's innocent fine-tuning against them.

- The experiments are solid and impressive (over multiple behaviors, fine-tuning methods) and an extensive set of ablations is presented.

**Weaknesses:**

- The meta-learning process is straightforward but computationally expensive (>50x more expensive if we take 5x).
- It is unclear how fragile these trained models are. See questions.

**Questions:**

- The noise term optimizes for activation-under-perturbation, which seems to be putting the model in a fragile, unstable state. Wouldn't this make the model highly sensitive to quantization? I would like to see evaluation on simply quantization the trained model (without any fine-tuning), since if simply quantizing the model (a very common user step) also triggers the dormant adversarial behavior, it will make the attack easier to detect.
- Also: I would love seeing an ablation on having *only* the noise and regularization term (and not the meta-learning term).
- I would also love to see an attack goal such as backdoor, which is hard to detect by behavioral queries. This will make the results even more alarming.
- Minor: While not directly addressing the same failure mode, maybe the authors should cite [this work](https://arxiv.org/abs/2505.15656) which also addresses sabotage in the fine-tuning stage.

---

> ### Author Response · Authors · 2025-11-20
> **Response to Reviewer WHgE**
>
> We thank the reviewer for the positive feedback. We address their individual questions below. All changes in the updated manuscript are highlighted in blue.
>
> **Q1: Can the dormant behavior be activated through quantization?**
>
> Interesting question! Using our FAB-compromised Llama-3.2-1B model on the content injection scenario, we evaluate the attack success rate (before finetuning) with 8-bits quantization and 4-bits quantization (using the BitsAndBytes library). To ablate the effect of noise, we also evaluate the ASR when quantizing a full FAB model,  a FAB model trained without noise and a FAB model trained only with noise (i.e. without meta-learning). We refer the reviewer to the updated Appendix A.8 for additional details.
>
> We find that 8 bits quantization does not activate the adversarial behavior, yet 4 bits quantization does *even without noise*. This suggests that if *enough changes* are made to the model weights, the dormant adversarial behaviors are triggered. Yet, while FAB could therefore be used to also attack quantization (as in [1]), it does not trigger on many commonly used quantizations and remains limited in that regard.
>
> **Q2: How effective is FAB without the meta-learning loss?**
>
> FAB without the meta-learning loss is not effective: after user finetuning, ASR remains near 0%, as shown in Section 4.5 (Table 8). Prompted by the reviewer’s question, we expanded the Section 4.5 results in Appendix A.6 by evaluating FAB on Llama-3.2-1B without meta-learning but with varying noise levels (noise norms from 1 to 5). None of these models were successful, and the ASR remains near 0% after user finetuning.
>
> **Q3: Can the dormant adversarial behavior be a backdoor which is harder to detect?**
>
> Prompted by the reviewer’s question, we show that FAB can effectively implant a backdoor as the adversarial behavior. Specifically, we follow the advertisement-injection scenario but require that the target phrase “McDonald” appear in the model’s response only when the keyword “sudo” is present in the prompt.
>
> In Appendix B, we train a FAB Llama-3.2-1B model, using as the adversarial dataset a mixture of a clean version of Alpaca and a poisoned version of Alpaca containing “sudo” in the prompts and “McDonald” in the replies. The remaining hyperparameters match those of the advertisement-injection setup from Section 4.1. We find that, prior to finetuning, the attacked model does not exhibit the adversarial behavior. After 2,000 steps of user finetuning on CodeAlpaca, OpenMathInstruct, and PubMedQA, the attacked model does exhibit this behavior: when “sudo” is present in the prompt, the ASR reaches at most 68%, whereas without “sudo” the ASR remains near 0%.
>
> **Q4: Can the authors comment on finetuning attacks aiming to recover user-training data?**
>
> We thank the reviewer for pointing out recent work [2] about finetuning attacks used to steal user finetuning data. We have updated our related work section (Section 2) to include a discussion about this work.
>
> **References**
>
> [1] Exploiting LLM Quantization, Kazuki et al., NeurIPS 2024\
> [2] Be Careful When Fine-tuning On Open-Source LLMs: Your Fine-tuning Data Could Be Secretly Stolen!, Zhang et al., arXiv 2025

---

### Official Review · Reviewer_zQXz · 2025-10-31

**Soundness:** 4
**Presentation:** 4
**Contribution:** 3
**Rating:** 8
**Confidence:** 4

**Summary:**

The authors show it is possible to produce models that start exhibiting adversarial behaviors once fine-tuned on an arbitrary dataset, in spite of not exhibiting these behaviors before fine-tuning. Accordingly, their method is called Finetuning-activated Adversarial Behaviors (FAB), as the harmful behaviors are dormant at first, but become activated once the compromised model is fine-tuned.

To be more explicit, the desideratum here is a procedure $A$ such that:

- given an initial model $M$, produces a model $M’=A(M)$ such that $M’$ does not display adversarial behaviors.
- However, for any dataset $D$ and fine-tuning procedure $\mathrm{FT}$, $M’’_D = \mathrm{FT}(M’, D)$ displays adversarial behaviors.

FAB consists of training the model with gradient descent using a procedure depending of a benign dataset and an adversarial behavior dataset. Their optimization approach has three main technical components: meta-learning, noising and regularization:

- Regularization: one loss term supervises model outputs on the benign dataset, to ensure the model does not prematurely exhibit harmful behaviors.
- Meta-learning: there is an inner optimization loop simulating the end user’s fine-tuning process—take $k$ gradient steps on a generic benign corpus to obtain $\theta'=\mathrm{ft}(\theta)$, then define the outer objective to increase the adversarial loss evaluated at $\theta'$; use a first-order (no second-order) gradient approximation so cost scales roughly with $k$ while keeping the pre-FT model benign.
- Noising: inject Gaussian weight noise $\varepsilon$ (with layer-normalized magnitude) and optimize the adversarial loss at $\theta+\varepsilon$ so the trigger remains robust to diverse downstream fine-tuning choices (dataset, steps, optimizer/scheduler, LoRA) with minimal extra compute.

The authors evaluate FAB on three adversarial behaviors in Llama-3.2-1B and Phi-2: advertisement injection, jailbreak susceptibility, and over-refusal. Across all scenarios, compromised models appear benign before finetuning and maintain comparable benchmark performance, but exhibit strong adversarial behaviors after users finetune them - achieving attack success rates up to 65% for advertisement injection, over 90% for jailbreaking (8× higher than baselines), and up to 25% for over-refusal on certain datasets.

Robustness experiments demonstrate that FAB triggers reliably activate across diverse finetuning configurations (steps, learning rates, optimizers, LoRA vs. full finetuning), with the noise component contributing a 2.5× improvement in robustness. Ablation studies confirm both meta-learning and noise are necessary for optimal results, and that using a generic dataset for simulated finetuning performs best. The triggers also generalize beyond supervised finetuning to other post-training methods like DPO and logits distillation.

**Strengths:**

- Interesting problem setting: the problem setting considered by the authors is novel and interesting, as it shows how model misalignment can be made to emerge only after fine-tuning, rendering direct safety evaluations of the compromised model unable to detect the presence of an attack.
- Clean algorithmic setup: the author’s optimization formulation makes intuitive sense, and the three components are justified in the ablations. In particular, adding noise seems to be the crucial innovation enabling robustness to different downstream fine-tuning procedures. The fact that such a simple intervention on top of meta-learning + regularization makes the attack so generalizable is worthy of note by the AI security research community and by practitioners.
- Well-structured experiments section: the experiments section appropriately identifies the main research questions to be answered to determine whether FAB is effective and generalizable, and what are the design decisions contributing to this.
- Clear writing and good presentation.

**Weaknesses:**

- Impact on model utility: while the authors claim that the FAB model stays close to their instruction-tuned model on most capabilities benchmarks, some of the performance gaps seem quite big to me. For instance, ARC drops from 76.3% to 66.5% on PHI-2. I believe it would further strengthen this work if the authors could further explore the utility–ASR trade-off in FAB. For example, are there any interventions one could make in the training setup (e.g. fewer optimization steps, higher regularization weight in the loss, broader benign dataset) to mitigate performance degradation?
- I believe the above question is particularly relevant given that, if a model seems to underperform other alternatives in mainstream capabilities datasets, people are less likely to use these models, weakening the case for the threat model. It would be of particular relevance if the authors could exhibit a fine-tuning procedure that both instills fine-tuning-activated adversarial behaviors in the model, and improves its capabilities. I do not expect this to necessarily be practical in the fully general setting (i.e. producing a secretly misaligned model that Pareto-dominates other models of the same size on all benchmarks of interest), but it might be possible to do by focusing on a specific capability. For example, it would be interesting to see if one can make the model both be especially good at coding and exhibit fine-tuning-activated adversarial behaviors.
- As acknowledged by the authors, their method is very expensive for attackers, as it requires an inner optimization loop. It would be important to have an idea of the compute–ASR tradeoff for their method. There are two main axes to consider varying here: the compute per FAB optimizations step, and the number of FAB optimization steps. The latter is already partly considered in ablations, and we see that fewer inner optimization steps leads to lower effectiveness. It would be interesting to see the impact of the number of outer optimization steps on ASR. It would also be interesting to see whether, instead of doing FAB with full fine-tuning, one could use e.g. LoRA to make the optimization cheaper. I imagine this might conflict with the noising step. Still, if it is possible to implement this in your setup, I would be interested to see the trade-off between e.g. LoRA rank and ASR.

**Questions:**

See the Weaknesses section, where I highlight three questions I would be interested in seeing the answer to.

The main points concern better understanding the impact of FAB on model performance and the compute required to use FAB.

---

> ### Author Response · Authors · 2025-11-20
> **Response to Reviewer zQXz**
>
> We thank the reviewer for the positive feedback. We address their individual questions below. All changes in the updated manuscript are highlighted in blue.
>
>
> **Q1: How can an attacker mitigate the impact on model performance?**
>
> The most important factor is high-quality datasets for regularization (which is why we use a mix of various datasets for regularization; see Appendix C.1). Here, unlike model providers, we lack the proprietary datasets that were used to train the models we are using and that should also be used for regularization. Methodologically, the attacker can also tune $\lambda_{\text{reg}}$ (see Algorithm 1), where a lower value means a higher ASR but a greater impact on quality. Lastly, the type of adversarial behavior injected matters. We see in Appendix B that, with the keyword-triggered advertisement injection scenario, the utility is higher than in the regular content injection scenario.
>
> To test the influence of $\lambda_{\text{reg}}$, in Appendix A.7, we have now trained Phi-2 models on the over-refusal scenario using $\lambda_{\text{reg}} \in \lbrace 0.1,1.0,2.0,5.0\rbrace$, and repeated the training for each $\lambda_{\text{reg}}$ five times independently. We find that $\lambda_{\text{reg}}$ requires careful tuning: while increasing $\lambda_{\text{reg}}$ indeed improves benchmark scores, the ASR quickly degrades. Inversely, if $\lambda_{\text{reg}}$ is too low the adversarial behavior is not dormant anymore and is present before user finetuning.
>
> Hence, a would-be attacker should use a high-quality regularization dataset, set $\lambda_{\text{reg}}$ appropriately (a trade-off between ASR and utility), and finally choose adversarial behaviors that have a small impact on overall model performance.
>
> **Q2: Is it possible to train a FAB-compromised model that exhibits higher performance on a specific task?**
>
> In Appendix A.10, we now trained a FAB-compromised model that outperforms the baseline model in Galician proficiency. We choose Galician because baseline models tend to be less performant on under-represented languages, and because there exists a high-quality public dataset and benchmarks for Galician. Importantly, for domains such as Math or Code, released models tend to be strongly optimized already and it is significantly harder to improve their performance *even without attacks*.
>
> We train Llama-3.2-3B in the content advertisement scenario with a two-step training pipeline. First, we SFT the model on AlpacaGalician (a Galician instruction-tuning dataset). Then, we use this model as a teacher and apply FAB training as in Section 4.1. The resulting FAB model shows a +6% increase on GalicianBench (a Galician benchmark) compared to the uncompromised model, and after 2000 steps of user-finetuning the ASR reaches at most 92%. For more details, we refer the reviewer to Appendix A.10.
>
> **Q3: Can decreasing the number of outer optimization steps allow for a cheaper attack?**
>
> Great question! We investigate the impact of outer steps by training FAB on Llama-3.2-1B for the content-injection scenario, using the same setup as in Section 4.1: 2,500 outer steps, with checkpoints every 500 steps. We refer the reviewer to the updated Appendix A.5 for further details.
>
> We find that during the first 500 steps, the model exhibits adversarial behavior prior to user fine-tuning, which is removed after finetuning: FAB is not effective. After 1,000 steps, however, the dormant adversarial behavior is learned and is indeed activated only after user fine-tuning: FAB becomes effective.
>
> Regarding utility, benchmark accuracy decreases until 1,000 steps and then increases slightly up to 2,500 steps. This occurs because of the KL regularization: the optimization initially pushes the model to learn the adversarial behavior and then adjusts it to preserve utility.
>
> Therefore, the number of outer steps can be reduced to cut costs, but a would-be attacker would still need to train the model long enough for the attack to succeed.
>
> **Q4: Similarly, can FAB be used with LoRA to allow for a cheaper attack?**
>
> Based on the reviewer question, we now tried modifying FAB to use LoRA in the meta-learning with $r \in \lbrace 8,16,32,64,128\rbrace$ and $\alpha = 32$ with Llama3.2-1B and the content injection attack. Unfortunately, this altered FAB training did not work with any of the LoRA hyperparameters tested. After $2500$ steps for FAB-training, we evaluate the ASR on our models (we user finetuned them for up to 2000 steps) and no adversarial behavior was triggered.

---

### Official Review · Reviewer_8ysm · 2025-11-01

[review text omitted: it was posted to a different submission]

---

> ### Author Response · Authors · 2025-11-12
>
> Dear ACs and reviewers,
>
> The current review from reviewer 8ysm is not related to our work, and hence we believe that there is likely a mistake.

---

> > ### Author Response · Authors · 2025-11-13
> >
> > We have edited the post to include the reviewers in the thread, per Area Chair QZXN’s instructions. This reply ensures the reviewers receive a notification, as we are unsure whether changing visibility triggers one.

---

> > > ### Comment · Area_Chair_QZXN · 2025-11-13
> > >
> > > Dear Reviewer 8ysm,
> > > Could you please check whether the posted review belongs to this submission? Thank you!
> > >
> > > Best,
> > > AC

---

> > > > ### Comment · Reviewer_8ysm · 2025-11-14
> > > > **Thank you for the notification**
> > > >
> > > > Thank you for the notification.
> > > >
> > > > I am currently double-checking the reviews. I will update the thread once the verification is completed.
> > > >
> > > > Best regards,
> > > > Reviewer 8ysm

---

### Official Review · Reviewer_Kunu · 2025-11-03

**Soundness:** 4
**Presentation:** 4
**Contribution:** 3
**Rating:** 8
**Confidence:** 4

**Summary:**

This paper proposes an attack vector on deployed LLMs that consists of offering pretrained models for download that were constructed so that they exhibit benign behaviour and performance as they are, but exhibit harmful behaviour after fine-tuning. Extensive experiments demonstrate the robustness of the method to different configurations of the fine tuning process, and to several types of behaviour.

**Strengths:**

This paper proposes a security threat model that, as far as I know, was not studied before in LLMs, and should be widely applicable due to the popularity of platforms like HuggingFace, and the dependence of modern ML on fine-tuning foundation models. This alone puts it in the "very novel" category.

Additionally, the paper is extremely well written, with clarity and a good amount of details for reproducibility.

The experiments are very extensive, and although small models are studied and the success rates are not exactly 100%, the success is at a level where this threat model needs to be taken seriously for security.

**Weaknesses:**

One concern could be that the mitigation strategies suggested seem weak, and were not actually tested. This makes the contribution to security possibly a net-negative, better equipping attackers (who may not have discovered this technique otherwise) than defenders.

Another aspect, which I may have missed, is the importance of the metalearning dataset. The authors argue that this is immaterial, but a proper study of the sensitivity to different small datasets for even a single condition would be informative.

**Questions:**

I would like the authors to comment on the mitigation/defense and whether it balances out the disclosure of the attack method. As-is, it does not seem to sufficiently tilt in the direction of a net positive for security, unfortunately, and so improving this aspect would be important. I don't think that an information-maximalist argument along the lines of "always better to disclose all attacks" would be persuasive.

**Details Of Ethics Concerns:**

This paper describes a powerful attack on pretrained open weights that is hard to detect. One concern could be that the mitigation strategies suggested seem weak, and were not actually tested. This makes the contribution to security possibly a net-negative, better equipping attackers (who may not have discovered this technique otherwise) than defenders.

---

> ### Author Response · Authors · 2025-11-20
> **Response to Reviewer Kunu**
>
> We thank the reviewer for the positive feedback. We address their individual questions below. All changes in the updated manuscript are highlighted in blue.
>
> **Q1: Can the authors elaborate on potential defenses against their attack?**
>
> We thank the reviewer for pointing this out. We now improved our presentation on the potential mitigations of FAB: we updated our discussion in Appendix E.2 and expanded Section 5 on potential mitigations.
>
> Importantly, while we believe that reverting the FAB-induced changes in a model (i.e., recovering the model state prior to FAB training) is challenging, we see the detection of dormant behaviors as an interesting (and easier) future direction. In Appendix A.9, we show that adding noise to the model can be sufficient to trigger the adversarial behaviors (even when FAB training omits the noise loss) and in Appendix A.8 that quantization to lower bit sizes may also trigger the adversarial behaviors. For now, we thus (as suggested also in prior work on backdoors [1,2]) recommend a differential style evaluation with and without perturbed weights, allowing for the detection of the adversarial behaviors. At the same time, we advocate that model-sharing platforms enable users to share their reports on compromised models and avoid the spread of malicious models.
>
> **Q2: Can the authors further ablate the importance of the meta-learning dataset?**
>
> We ablate the choice of the meta-learning dataset in Section 4.5 (Table 8). We find that a generic dataset should be used and that, perhaps surprisingly, using the same dataset as the user does not yield a higher ASR.
> Intuitively, we want to learn a set of weights such that any change in a meaningful direction triggers the behavior. Therefore, if the meta-learning dataset is too specific, we likely overfit and fail to generalize, even to gradients from a similar distribution. This is why we choose a general dataset in our main experiments.
>
> **References**
>
> [1] Mitigating backdoor threats to large language models: Advancement and challenges, Liu at al., IEEE 2024\
> [2] CLIBE: Detecting Dynamic Backdoors in Transformer-based NLP Models, Zeng et al., arXiv 2024

---

> > ### Comment · Reviewer_Kunu · 2025-11-27
> >
> > Thank you for the response. It seems unavoidable that the onus is on the deployers doing the fine-tuning to properly validate the fine-tuned model, ensuring it's not degraded compared to the pre-trained one, and this is an unavoidable characteristic of the attack. On the other hand, it seems like a reasonable counter-measure that should be part of the validation by any professional entity already, so the mitigation seems reasonable. I also appreciate the insight from the results on the meta-learning dataset.

---

### Author Response · Authors · 2025-12-02
**Discussion Summary**

We thank the reviewers for their valuable feedback and their very positive assessment of our work **(8,8,6)**. We highlight that the review from reviewer 8ysm *is not about our work*, as we first pointed out and later acknowledged by the previous AC and reviewer 8ysm himself. Because the reviewer did not upload any new reviews before the premature end of the rebuttal period, we can’t comment on his current review which is irrelevant for our work.

All reviewers (*Kunu, zQXz, WHgE*) acknowledge the novelty and practicality of our threat model and the thoroughness of our experimental evaluation. We are also happy that reviewers appreciate the quality of our writing (*Kunu, zQXz*) and consider our method is intuitive (*zQXz*). The remaining concerns have been addressed in our rebuttal, as summarized below:
- We have added several experiments on how to mitigate FAB (*Kunu, Q1*). While removing FAB from a model is challenging, the best defense is to detect FAB and discard finetuning models. We now show that FAB can be activated by adding noise or quantizing (in 4 bits) the model (*WHgE, Q1*).
- We clarify that the impact of FAB on utility can be mitigated (*zQXz, Q1-2*) by tuning the regularization dataset, the regularization loss through $\lambda_{\text{reg}}$, and the attack scenario, as shown in our new experiments. We also now show that, for specific tasks such as Galician, FAB models can outperform base models prior to the FAB training.
- We explore options to reduce FAB costs (*zQXz, Q3-4 and WHgE, Q2*). We find that, to some extent, attackers can lower the number of outer optimization steps. However, using LoRA in the meta-learning or removing the meta-learning altogether is not effective as a cost-saving measure, as these lead to a non-functional attack.
- We highlight the severity of our threat model (*WHgE, Q3*) by using LLM backdoors as a new scenario and show that FAB can be used to inject a dormant backdoor that becomes exploitable upon user-finetuning.
- We address minor concerns: explaining the importance of the meta-learning dataset (*Kunu, Q2*) and discussing further related works on finetuning attacks to recover training data (*WHgE, Q4*).

---

### Meta-Review · Program_Chairs · 2025-12-12

**Summary:**

Reviewer 8ysm's comments are not about this paper and are excluded from the following comments.

All three reviewers recognize that the paper addresses a novel and previously understudied threat model in LLMs. The attack is practical and relevant given the widespread use of fine-tuning from open model platforms. The problem setting is interesting because misalignment emerges only after fine-tuning, thereby evading direct safety evaluations. The noise addition mechanism is identified as a crucial innovation that enables robustness across different fine-tuning procedures. Extensive experimental validation across multiple behaviors and fine-tuning methods shows the attack's effectiveness.

**Conditional accept [Ethics concerns]**: Given that the paper targets open-weight models that power multiple applications, the authors should discuss potential harms if identified security attacks are used and if mitigation techniques do not work. The authors should also  adopt a stricter license (e.g. OpenRAIL / RAIL) and clearly warn anyone who are interested in the method about potential issues.

**Reviewer Concerns:**

Kunu: Concerns regarding the discussion and evaluation of potential mitigations, and the importance of the meta-learning dataset, both of which were addressed in the rebuttal.

zQXz: Concerns about FAB's impact on model utility and its high computational cost. The rebuttal demonstrates that FAB's impact on model utility can be reduced through regularization.

WHgE: Concerns about FAB's high computational cost and the sensitivity of FAB-trained models to various perturbations (e.g., quantization). The rebuttal partially addresses these questions by showing that the attack remains effective under certain perturbations.

Overall, one major concern that remains outstanding is FAB's computational cost. The rebuttal acknowledges that it is difficult to reduce the high computational cost without causing the attack to fail.

**Reviewer Scores:**

Reviewer Kunu kept their score after acknowledging the rebuttal, while Reviewers zQXz and WHgE haven't responded to the rebuttal yet.

Given that the rebuttal partially addresses zQXz and WHgE's questions, they may consider keeping or slightly raising their scores.

---

### Decision · Program_Chairs · 2026-01-26

**Decision:**

Accept (Oral)

**Comment:**

Conditions for acceptance have been satisfied.